# Anterior Cruciate Ligament Reconstruction with Modified Transtibial Technique: Outcomes and Return to Sport in Athletes

**DOI:** 10.3390/healthcare13091056

**Published:** 2025-05-04

**Authors:** Arcangelo Russo, Giuseppe Gianluca Costa, Maria Agata Musumeci, Michele Giancani, Calogero Di Naro, Francesco Pegreffi, Gianluca Testa, Marco Sapienza, Vito Pavone

**Affiliations:** 1Faculty of Medicine and Surgery, Kore University of Enna, 94100 Enna, Italy; arcangelorusso@me.com (A.R.); gianlucacosta@hotmail.it (G.G.C.); calogerodinaro@gmail.com (C.D.N.); francesco.pegreffi@unikore.it (F.P.); 2Orthopaedic and Traumatologic Unit, Umberto I Hospital, Azienda Sanitaria Provinciale di Enna, 94100 Enna, Italy; giancani63@gmail.com; 3Department of General Surgery and Medical Surgical Specialties, Section of Orthopedics and Traumatology, A.O.U. Policlinico Rodolico-San Marco, University of Catania, 95100 Catania, Italy; musumeci.maria.mm@gmail.com (M.A.M.); gianpavel@hotmail.com (G.T.); vitopavone@hotmail.com (V.P.); 4Department of Precision Medicine in Medical, Surgical and Critical Care (Me.Pre.C.C.), University of Palermo, 90133 Palermo, Italy; 5Unit of Recovery and Functional Rehabilitation, Umberto I Hospital, Azienda Sanitaria Provinciale di Enna, 94100 Enna, Italy

**Keywords:** anterior cruciate ligament, return-to-sport, failure, athletes, arthrometer, KT-1000

## Abstract

Background: Anterior cruciate ligament (ACL) injuries are common among athletes and significantly impact their knee stability and performance. Surgical reconstruction is the standard treatment. The modified transtibial technique has emerged as a promising surgical approach for optimal graft positioning and complication reduction. Methods: A retrospective study of athletes who underwent primary ACL reconstruction with the modified transtibial technique was conducted. Clinical outcomes were evaluated using the Lysholm and International Knee Documentation Committee (IKDC) subjective scores and objective knee stability assessments. Return-to-sport rates and associated factors were analyzed. Results: Forty-four athletes were included (thirty-seven males, seven females; mean age 21.2 ± 5.0 years). At mean follow-up of 27.0 ± 12.2 months, significant improvements in the Lysholm and IKDC subjective scores were observed. Overall, 88.2% of athletes returned to sports, and 65.9% achieved their pre-injury levels. Return to pre-injury level was defined as regaining the same type, intensity, and frequency of sport participation as before the injury occurred. Professional athletes showed significantly higher return-to-pre-injury-sport rates (79.3%) than recreational athletes (40.0%, *p* = 0.0091). Concomitant meniscus injuries negatively impacted return-to-sport rates (92.9% versus 66.7%, *p* = 0.0397). The overall failure rate was 4.6% (2/44; 95% confidence level [CI]: 0.6–15.5%) with two cases of graft insufficiency or re-rupture. Conclusions: ACL reconstruction with the modified transtibial technique provides favorable clinical outcomes, high return-to-sport rates, and low failure rates, particularly among professional athletes. Meniscus preservation is crucial for optimizing post-operative recovery. Future research should focus on long-term outcomes and comparative studies with other ACL reconstruction techniques.

## 1. Introduction

The anterior cruciate ligament (ACL) is crucial for knee stability, especially in sports that require dynamic movements, such as pivoting, cutting, and jumping [1]. ACL injuries are particularly common among athletes, both recreational and professional, and can significantly impair their performance and quality of life [2,3].

Surgical treatment of ACL injuries is supported by most authors for restoring joint kinematics, preserving the cartilage and meniscus structures, and increasing the chances of returning to sport activities [2]. Anatomic ACL reconstruction is universally recognized as “the gold-standard” [2]. The use of this type of reconstruction is driven by growing recognition of the limitations of the traditional transtibial technique, which often results in non-anatomic femoral tunnel placement due to its dependence on the tibial tunnel trajectory. This suboptimal positioning has been associated with persistent rotational instability and higher rates of graft failure. In contrast, independent femoral tunnel drilling techniques, such as the anteromedial (AM) portal and outside-in approaches, allow for more precise placement of the femoral tunnel at the native ACL footprint, a process that leads to improved rotational stability and kinematic restoration [4,5]. Consequently, these techniques have become preferred in contemporary ACL reconstruction. However, recent modifications of the traditional transtibial technique made this approach more accurate by providing precise graft placement and minimizing surgical complications [1]. In contrast to the conventional transtibial technique, which often results in a more vertical and non-anatomic femoral tunnel due to its dependence on the tibial tunnel trajectory, the modified transtibial technique incorporates adjustments in tibial tunnel orientation and drilling angles to facilitate a more anatomical femoral tunnel placement. By optimizing tunnel alignment, this approach aims to more accurately replicate the native ACL footprint on the femoral side, thereby enhancing rotational stability and restoring more physiological knee kinematics [5,6,7]. Theoretical advantages of this technique include a suboptimal graft orientation with lower bending angle [6], which may lead to a reduction in the stress on the graft, thus facilitating graft incorporation and longevity [7].

Several studies describe the outcomes of the modified transtibial technique in terms of knee stability and function [1], although outcomes can vary based on athlete demographics and injury severity [8,9]. Questions remain regarding the results of the modified transtibial technique and whether the results indicate achievement of both favorable clinical outcomes and successful return-to-sport, particularly when comparing recreational and professional athletes. The return-to-sport and current level of activity are not secondary issues since a stable knee may not be good enough for the athletic demands of elite-level sports in cases of lower game performance [3,8].

Despite significant advancements in surgical techniques, considerable variability in clinical outcomes following ACL reconstruction remains. One of the key unresolved issues is the influence of femoral tunnel positioning and graft orientation on knee stability, graft longevity, and return-to-sport rates. While anatomic reconstruction has become the gold standard, the specific biomechanical and clinical implications of different femoral drilling techniques are still debated [1,2,3,4,5,6,7]. We hypothesized that ACL reconstruction using the modified transtibial technique, which allows for more anatomic femoral tunnel placement, would demonstrate superior rotational stability as measured by pivot shift testing and higher rates of return to pre-injury sports levels compared to those with non-anatomic tunnel placement”.

The purpose of this study was to evaluate clinical outcomes and return to sport after ACL reconstruction with a modified transtibial technique. The hypothesis states that good clinical outcomes and high rates of return-to-sport activities, both in professional and recreational athletes, can be achieved using this technique,

## 2. Materials and Methods

A retrospective analysis of all sport practitioners undergoing primary ACL reconstruction between January 2020 and December 2023 was conducted according to the Strengthening the Reporting of Observational Studies in Epidemiology (STROBE) guidelines [10]. The study received the local ethics committee approval (no. 215/CEL in 21 October 2024) prior to data extraction and was performed in accordance with the World Medical Association’s 1964 Declaration of Helsinki and its later amendments [11]. All patients included in this study signed specific informed consent forms for data collection and elaboration in an anonymous and/or aggregate form.

### 2.1. Patient Inclusion and Selection Criteria

Inclusion criteria included several parameters: (1) primary ACL reconstruction during the timeframe reported above, (2) patients participating in team or individual sports and those who require regular competition and constant training, (3) individuals consenting to personal data collection for the purpose of the study, and (4) minimum follow-up of 12 months. A patient was defined as a professional athlete if they obtained more than 50% of his income through his/her sport; otherwise, they were considered a recreational athlete. Recreational athletes were defined as individuals participating in sports primarily for leisure or fitness, typically without regular competitive involvement at a high level.

After identification of all eligible cases, a stepwise exclusion process was followed if any one of several exclusion criteria was fulfilled: (1) concomitant procedures requiring longer period of rehabilitation and longer absence from sport activity (posterior cruciate ligament reconstruction, chondral or osteochondral transfer surgery, osteotomies); (2) past medical history of ACL injuries in the contralateral knee before the index surgery; (3) locked knees for articular loose bodies or displaced bucket-handle meniscus tears, which prevent proper evaluation of knee stability; (4) patients undergoing a second or subsequent ACL reconstruction on the same knee as their rehabilitation and outcomes may differ from primary reconstructions; and (5) individuals with a history of significant surgical procedures on the same knee (such as meniscectomy, cartilage repair), which could affect post-operative recovery and study outcomes.

A formal a priori power analysis was not conducted due to the retrospective nature of the study and limited pool of eligible subjects. From an initial cohort of 62 competitive athletes who fulfilled the inclusion criteria, 44 were available and included in the final analysis. Despite the absence of an a priori calculation, this sample size was considered adequate for exploring clinically meaningful differences in functional outcomes, graft survival, and return-to-sport rates, given the homogeneity of the population and the comprehensive clinical, instrumental, and surgical data that were collected.

### 2.2. Surgical Technique and Rehabilitation Protocol

In all patients, ACL reconstruction was performed using the same modified transtibial technique [12] as shown in (Figure 1). The tibial tunnel was created by identifying the entry point on the tibial cortex just medial to the tibial tubercle near the anterior fibers of the medial collateral ligament. The tibial guide was set at an angle of 45°, which allowed for a more posterior and medial orientation of the tunnel compared to the conventional technique, which facilitates a more anatomic placement of the femoral tunnel. An offset femoral guide was then inserted through the tibial tunnel and advanced toward the lateral femoral condyle. Under direct visualization, the guide was gently rotated until it reached the center of the native femoral ACL footprint, typically at the 10 o’clock position in right knees and 2 o’clock in left knees. This maneuver allows for anatomic femoral tunnel positioning while preserving the ease and reproducibility of the transtibial approach. Following this step, the femoral tunnel was drilled, and the graft was pulled intra-articularly through the tibial tunnel. Femoral fixation was achieved using a cortical suspension device (Endobutton, Smith & Nephew, Andover, MA, USA). On the tibial side, fixation was performed at 20° of knee flexion. For soft-tissue grafts, a bioabsorbable interference screw (BIO-SURE-HA, Smith & Nephew) was used to minimize long-term foreign body presence and facilitate biological integration. In cases involving bone blocks, such as with bone–patellar tendon–bone (BPTB) or quadriceps tendon (QT) grafts, a 7 mm metallic interference screw (SOFT SILK, Smith & Nephew) was selected to ensure optimal fixation strength and stability at the bone interface. Autograft choice hamstrings, BPTB, or QT with bone block were guided by a combination of surgeon experience and patient-specific factors, including anatomical considerations, graft availability, previous injuries, and the type of sport the patient aimed to return to. For example, BPTB grafts were often favored for athletes participating in high-demand pivoting sports due to these grafts’ superior bone-to-bone healing properties, while hamstring grafts were considered in cases in which reduced anterior knee pain and/or faster recovery was prioritized. QT grafts offered a versatile alternative with favorable biomechanical strength and were particularly useful in revision settings or when previous grafts had been used. In selected cases, a lateral extra-articular tenodesis (LET) was performed using the Cocker–Arnold technique [13]. This additional procedure was indicated in specific cases, including those with a graft diameter < 8 mm (an established threshold associated with increased risk of graft failure, particularly in young and active patients [13]), high-grade pivot shift (Grade III) without lateral meniscus injury, patient age under 30 years, involvement in high-demand sports, and/or clinical signs of constitutional ligamentous laxity [13,14]. However, the indications for its use remain somewhat subjective, which may introduce variability in the interpretation of its effectiveness. Post-operatively, all patients were fitted with a hinged knee brace that was locked in full extension during ambulation and sleeping for the first three to six weeks. The variation depended on the graft type and any concomitant procedures that had been performed. For example, longer immobilization (closer to six weeks) was generally recommended in cases involving quadriceps tendon grafts or concurrent meniscal repairs to protect the healing structures. Range of motion exercises and full weight-bearing were progressively allowed between weeks three and six. The return-to-sport timeline was individualized based on patient progress, neuromuscular control, and completion of functional testing for which forward running and sport-specific drills were introduced at 10–12 weeks with full return to cutting and pivoting sports generally permitted after 5–6 months once quadricep strength and psychological readiness were fully recovered.

### 2.3. Data Collection

For each selected case, medical records and surgical reports were reviewed by three authors who extracted several sets of data: (1) patient’s sex, (2) age at the time of surgery, (3) body mass index (BMI), (4) time elapsed from the injury to the surgery, (5) data obtained from the standard pre-operative clinical evaluation (including the Lachman and pivot shift tests and arthrometric examination), (6) patient’s level of sports activity (rated according to the Tegner Score [14]), (7) pre-operative clinical scores (Lysholm and the International Knee Documentation Committee (IKDC) subjective scores [15,16]), and (8) details of the surgical procedure (including presence of associated medial meniscal lesions with related treatment, type of chosen graft with size, concomitant LET procedures).

The standard pre-operative clinical evaluation was performed using the objective IKDC form by two of the three most experienced surgeons in knee arthroscopy among the authors [17]. This evaluation was performed in the operating room for each case while the patient was under epidural or general anesthesia. The standard evaluation of knee stability included the Lachman and pivot shift tests. The Lachman test was scored as 0 (no increased laxity), +1 (slightly increased laxity with a firm end point), +2 (increased translation with a delayed end point), or +3 (translation with soft end point). A Lachman test result of 0 or +1 was defined as low grade, while a test rated +2 or + 3 was considered high grade. Similarly, the pivot shift test was scored as 0 (with no shift), 1+ (gliding), 2+ (clunk), or 3+ (gross rotary laxity). A 0 or +1 pivot shift test was categorized as low grade; otherwise, it was counted among high-grade tests. Any disagreement was discussed between the authors until a joint agreement was reached and transcribed in the medical record.

In addition to this, the KT-1000 knee ligament arthrometer (MEDmetric Corp, San Diego, CA, USA) was used to objectively quantify the anteroposterior laxity of the knee. [18] The test was performed by one of the same three experienced authors while the patient was awake in his room. The anteriorly directed push was applied to both knees using the manual-maximum force. The side-to-side difference was considered normal if the result was <3 mm, higher than usual if between 3 mm and 5 mm, and abnormal if >5 mm.

### 2.4. Final Follow-Up Examination

All selected athletes were recalled and invited for clinical examination after at least 12 months of follow-up. The clinical examination consisted of an evaluation of the outcomes using the same clinical and functional scores as described above (Lysholm and IKDC subjective scores) in addition to objective assessment of the knee stability using the Lachman, pivot shift, and arthrometric tests. These tests were administered by one of the same experienced knee surgeons who had evaluated the joint stability before the surgery. In addition to this step, athletes were asked if they had returned to sports, the type of resumed sport (categorized according to the Tegner Score), and time needed for full return-to-sport activity. Return to sport was defined as the resumption of any level of sports practitioner activity following surgery, while return to pre-injury level referred to regaining the same type, intensity, and frequency of sports participation as before the injury. The Tegner activity scale was used to assess and compare pre- and post-operative activity levels. In cases in which the athletes did not return to sports, the reasons for this lack of return were investigated.

If the athletes were not available for the clinical examination, clinical outcomes were assessed by phone interview to collect functional scores and data related to sports activity. Complications, re-operations with related reasons (including ACL graft revision), and ACL graft re-ruptures were also investigated and recorded. Contralateral ACL ruptures were also queried and reported. Overall composite failure rate was calculated at the final follow-up as the sum of graft re-ruptures (with or without ACL revision surgery) and clinical failures (graft insufficiencies demonstrated with at least two of the three following criteria: (1) high-grade Lachman test ≥ 2, (2) high-grade pivot shift test ≥ 2, and (3) abnormal KT-1000 side-to side difference > 5 mm.

### 2.5. Statistical Analysis

Statistical analyses were performed using SPSS^®^ Version 25 (SPSS Inc, Chicago, IL, USA). Categorical variables were reported as percentages or frequencies, and continuous variables were expressed as arithmetic mean ± standard deviation (SD). Comparison of the categorical variables was performed using the Pearson’s chi-squared test or Fisher’s exact test where appropriate. Comparison of the continuous variables was performed with the paired-samples T-test or the Mann–Whitney non-parametric test after examining the normality of data distribution using the Shapiro–Wilk test. Kaplan–Meier survival curves were generated to analyze survivorship using based on several endpoints: (1) cumulative ACL failure, (2) contralateral ACL rupture, and/or (3) re-operation for any reason except for ACL graft revision and contralateral ACL rupture. Comparison of survival curves between professional and recreational athletes was performed using the Mantel–Cox log-rank test. The level of significance was set at α = 0.05, and *p* values  <  0.05 indicated a statistically significant difference.

## 3. Results

From an initial count of 62 eligible athletes, 44 athletes were available and included in the present study. Of the latter, 29 (65.9%) were available for in-person clinical and instrumental evaluation, while the remaining 15 (34.1%) were evaluated by phone interview to collect functional scores and return to sports and to document any complications, re-ruptures, and revision surgeries. The final sample consisted of seven (15.9%) female and 37 (84.1%) male athletes with a mean age at surgery of 21.2 ± 5.0 years (Table 1).

Twenty-nine (65.9%) professional athletes (including 29 soccer players, 2 tennis players, 2 basketball players, and 1 football player) and 15 (34.1%) recreational athletes (11 soccer players, 2 volleyball players, 1 tennis player, and 1 martial artist) were included (Figure 2).

In most cases (40 patients, 90.9%), quadrupled hamstrings grafts were chosen for the ACL reconstruction, while BPTB and QT grafts were chosen for three (6.8%) and one (2.3%) patients, respectively. A total of 42 concomitant menisci tears were found in 30 (68.2%) of the patients. Among the latter, 8 patients had concomitant medial meniscus tears (7 ramp lesions, 1 bucket-handle tear of the medial meniscus), 10 patients had concomitant lateral meniscus tears (four posterior root tears, four longitudinal tears, and two horizontal tears), and 12 patients had concurrent medial and lateral meniscus tears (8 ramp lesions of the medial meniscus, 2 medial bucket-handle tears, 1 longitudinal medial meniscus tears, and 1 radial medial meniscus tears associated with eight posterior root tears of the lateral meniscus, two longitudinal tear of the lateral meniscus, and two tears of the popliteomeniscal fasciculi). Although concomitant meniscus tears were less common among professional athletes (69.0%) than recreational athletes (73.3%), this difference was not statistically significant (Figure 3).

In only 3 out of 42 cases, partial meniscectomy was the treatment of choice, while all remaining tears were treated using an all-inside or a transtibial pull-out technique (specifically for the posterior root tears of the lateral meniscus). A concomitant LET procedure was performed in three cases (6.8%).

At evaluation at the last follow-up after a mean of 27.0 ± 12.2 months, a significant improvement was reported from baseline in terms of both Lysholm score (from 79.3 ± 8.6 at baseline to 95.4 ± 5.8 at last follow-up; *p* < 0.0001) and IKDC subjective score (from 78.5 ± 8.6 at baseline to 91.2 ± 7.9 at last follow-up; *p* < 0.0001) as shown in Figure 4.

After the rehabilitation protocol, 39 athletes (88.6%) returned to sports after a mean of 8.1 ± 1.4 months (range 5–12 months). At the last follow-up, 33 athletes (75.0%) were still involved in sports activities of whom 29 athletes (65.9%) returned to the same pre-injury level. The return-to-sports rate was higher among professional athletes (82.8%) than among recreational athletes (60.0%), although this difference did not reach statistical significance (*p* = 0.0984). Similarly, return to the same pre-injury sport level was higher in professional athletes (79.3%) than in recreational athletes (40.0%), but in this case, a statistically significant intergroup difference was found (*p* = 0.0091). In ten out of eleven cases of failed return to sport, athletes had undergone concomitant meniscus treatment during the index ACL reconstruction. Indeed, athletes with isolated ACL lesions had higher chances of returning to sport (92.9%) compared to athletes with concomitant menisci tears (66.7%; *p* = 0.0397). Reasons for failed return to sport included fear of graft re-rupture in five cases, subsequent re-operation in two cases, subjective joint instability in two cases, knee swelling in one case, and job-related reasons in the last remaining case. The mean pre-injury Tegner score was 8.1 ± 1.2, while the mean post-operative Tegner score at final follow-up was 7.4 ± 1.5. Although a slight decrease was observed at the last follow-up, most patients (65.9%) returned to their pre-injury level or one below the pre-injury level.

The objective assessment at the last follow-up of the 29 athletes available showed good knee stability in most cases. High-grade Lachman test results in only two cases (6.9%) and high-grade pivot shift results in only one case (3.4%) were obtained. The same athlete had high-grade Lachman and pivot shift test results and was counted among clinical failures. Similarly, KT-1000 side-to-side improved from 5.4 ± 1.4 mm at baseline to 1.6 ± 1.4 mm (*p* < 0.0001). In 27 cases, the KT-1000 side-to-side difference was normal, in 1 case (3.4%) was higher than normal, and in the 1 remaining case (3.4%) was abnormal. This last patient was the same with abnormal Lachman and pivot shift test results. Return-to-sports rates and clinical examination data at last follow-up are summarized in Table 2.

Failure and re-operation rates (with subgroup analysis of reoperation for contralateral ACL ruptures and reoperation for any other reason) are shown in Figure 4 and summarized in Table 2.

Complications included one case of injury of the infrapatellar branch of the saphenous nerve, one case of deep infection, and one case of cyclops syndrome (these last two patients were reoperated for these reasons). The overall re-operation rate, excluding ACL revisions, was 20.5% (9/44; 95% confidence interval [CI]: 10.0–35.7%).

Four patients (two professional and two recreational athletes, 9.1%) underwent re-operations for failure of meniscal repairs (three patients for failure of ramp lesion repair, one patient for failure of both ramp lesion and posterior lateral meniscus root repairs), three patients (two professional and one recreational athlete, 6.8%) because of contralateral ACL rupture, and the last two professional athletes (4.6%) for deep infection and cyclops syndrome, respectively.

The composite failure rate was 4.6% (2/44; 95% CI: 0.6–15.5%). Failures included one professional soccer player with graft insufficiency at the last follow-up (demonstrated by high-grade Lachman and pivot shift test results and abnormal arthrometric values) and one professional athlete with graft re-rupture following a knee sprain during soccer match 17 months after surgery.

## 4. Discussion

The primary findings of this study indicate that ACL reconstruction using the modified transtibial technique in a cohort of both professional and recreational athletes leads to satisfactory clinical outcomes and low rates of failure.

While many of the findings, such as functional improvement and return-to-play (RTP) rates, have been well documented in the literature, our study specifically focuses on these parameters in athletes, thus making this specialized focus a key strength of our work. By examining these outcomes in an athletic population, we provide valuable insights into how specific factors, such as sport-specific demands and recovery timelines, may influence RTP in this unique cohort.

### 4.1. Graft Failure Rates

Both functional scores used in this study showed significant improvements in all cases in addition to exceeding the minimal clinically important difference described in the literature [19]. Furthermore, only two cases of failure were found, which corresponded to a rate of 4.6%. This failure rate is in line with comparisons to published benchmarks in the literature, and these results highlight that our observed failure rate (4.6%) falls within or below the expected range reported in similar populations undergoing ACL reconstruction [16,20]. This finding needs to also be interpreted carefully when considering the young age of the sample (21.2 ± 5.0 years), which is widely recognized as an independent factor for the risk of failure of ACL reconstruction [2,21]. Furthermore, this failure rate is extracted from a case series of ACL reconstructions with a low incidence of LET procedures (6.8%). Several studies recommend additional LET in high risk-athletes similar to those evaluated in the present article to reduce the risk of graft failure or residual rotational instability [22,23].

### 4.2. Associated Meniscal Lesions and Their Management

On the other hand, the high incidence of associated meniscal injuries and high prevalence of meniscal repairs in this series should be considered when discussing the low rate of graft failures and abnormal joint laxity. It is well known that the medial and lateral menisci significantly contribute to knee stability by acting as secondary restraints for translational and rotatory tibial displacement [2,24]. Meniscus repair seems to restore knee stability to levels that are comparable to ACL-reconstructed knees with intact menisci [25]. However, attempts to preserve meniscus function may result in higher risks of re-operation. Indeed, failed meniscus repairs represented the most common reason for re-operation in this series and accounted for two out of three cases of subsequent surgeries. Interestingly, the contralateral ACL injury rate was found to be higher than the ipsilateral graft rupture rate. Nonetheless, these data should not be unexpected as they confirm well-known findings in the literature [21,26]. Plausible explanations for this finding are young age [21,26], theoretical anatomical factors predisposing to ACL rupture [2,26], and great stresses to which the ACL is subject during high-demand sports activities [2,21].

### 4.3. Return to Sport (RTS)

Another important topic for discussion is the return-to-sport rate in this series and the level of resumed sport. A trend was found in favor of professional versus recreational athletes regarding return to sport, although this difference was not found to be statistically significant. Analyzing data more thoroughly, professional versus recreational athletes were more likely to return to pre-injury performances (79.3% versus 40.0%; *p* = 0.0091). Similar findings have already been reported in previous systematic reviews [27] and emphasize the prognostic role of the practiced level of sports. This discrepancy may be attributed to differences in rehabilitation adherence, access to specialized care, and psychological readiness, which are factors that have been extensively documented in the literature [28]. The use of “satisfactory outcomes” has been refined by reporting actual functional score values (such as Lysholm, IKDC, Tegner) and comparing them to established thresholds for clinical success or previously published studies [27,28]. Additionally, fear of re-injury emerged as a major barrier to the return-to-sport as five athletes explicitly cited that it was the reason for not returning, a result that supports previous studies that indicate psychological factors significantly impact the return-to-sport rates [29,30]. Interestingly, concurrent meniscal injuries were a negative prognostic factor for returning to sport and caused a reduction in the chance to return to sport by approximately -25%. These data confirm the non-secondary role of the menisci in knee kinematics [24] and should guide surgeons specializing in the care of sports trauma during clinical practice. Meniscus tears and partial meniscus resection have been clearly documented as a risk factor for delayed return to sport and career shortening in athletes [2]. Meniscus repair is supported by the authors of the present article, although a high risk of re-operation should be taken into consideration as the price to be paid.

### 4.4. Limitations

The findings of this study must be interpreted after considering several limitations. First, the retrospective design introduces a potential selection bias, although data collection was conducted according to standardized protocols. The mean follow-up period of 27 months is sufficient for evaluating short- to mid-term outcomes but does not allow assessment of long-term graft survival. No adjustments for multiple comparisons were applied, which may have led to an increase in the risk of type I errors in subgroup analyses and should be considered when interpreting the results.

Anterior tibial translation was quantified using arthrometric devices; however, rotatory knee stability was assessed manually without the aid of advanced tools, such as accelerometers. Despite this, manual evaluation remains the gold standard according to the literature and is included in the objective IKDC form [17].

Another limitation is the absence of a control group treated with an alternative ACL reconstruction technique (such as independent drilling), which limits a researcher’s ability to make direct comparisons in clinical outcomes and renders conclusions about the superiority of the modified transtibial technique speculative. Moreover, the inclusion of a small number of patients who received BPTB or quadriceps tendon grafts may have introduced variability and, thus, represent a potential source of bias. Similarly, the lack of stratification based on meniscal management (no tears, partial meniscectomy, or meniscus repairs) could have influenced return-to-sport outcomes, thus introducing further bias into the analysis. Outcome assessments were conducted using both in-person clinical evaluations and telephone interviews for follow-up (34.1% of cases) and, thus, introduced heterogeneity that may affect the consistency and reliability of the results, particularly in assessing knee stability.

Finally, the predominance of male soccer players in the study cohort introduces a population bias that may limit the generalizability of the results to female athletes, participants from other sports, or recreational populations.

Future studies should incorporate prospective randomized designs that compare modified transtibial and independent drilling techniques to allow researchers to draw more robust conclusions. Last, while psychological readiness was identified as a significant predictor of return to sport, additional investigations are needed to determine the role of sport-specific rehabilitation protocols and psychological support in optimizing recovery.

## 5. Conclusions

This study confirms that ACL reconstruction using the modified transtibial technique is an effective approach to restore knee stability and facilitates return to sport in competitive athletes. The technique resulted in significant functional improvements and a low failure rate across varying levels of athletic participation. Notably, the presence of concomitant meniscal injuries emerged as a relevant factor influencing R TP outcomes. While these findings support the clinical utility of this surgical approach, the retrospective design, gender imbalance, and partial reliance on phone-based follow-up highlight the need for caution in generalizing the results. Further prospective studies with longer follow-up periods and direct comparisons to alternative surgical techniques are warranted to refine patient selection criteria and optimize long-term outcomes.

## Figures and Tables

**Figure 1 healthcare-13-01056-f001:**
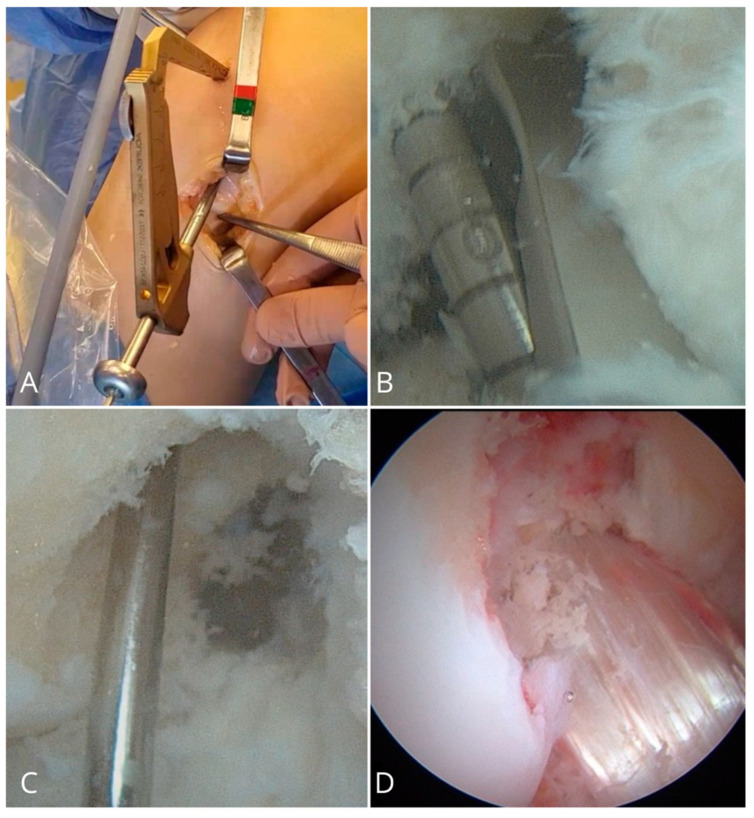
(**A**) Starting point of tibial tunnel, placed to create a more oblique femoral tunnel; (**B**) the creation of the tibia tunnel following the femural anterior cruciate ligament (ACL) footprint; (**C**) evaluation of the tibal tunnel; (**D**) positioning of the ACL graft.

**Figure 2 healthcare-13-01056-f002:**
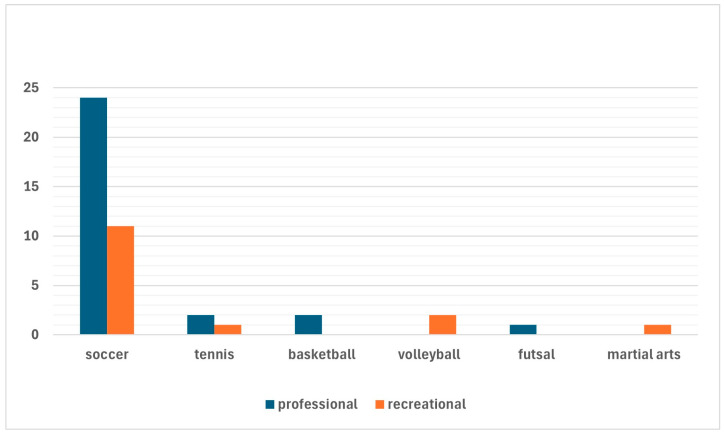
Sports activities performed by the athletes included in the present study with differentiation between professional and recreational athletes.

**Figure 3 healthcare-13-01056-f003:**
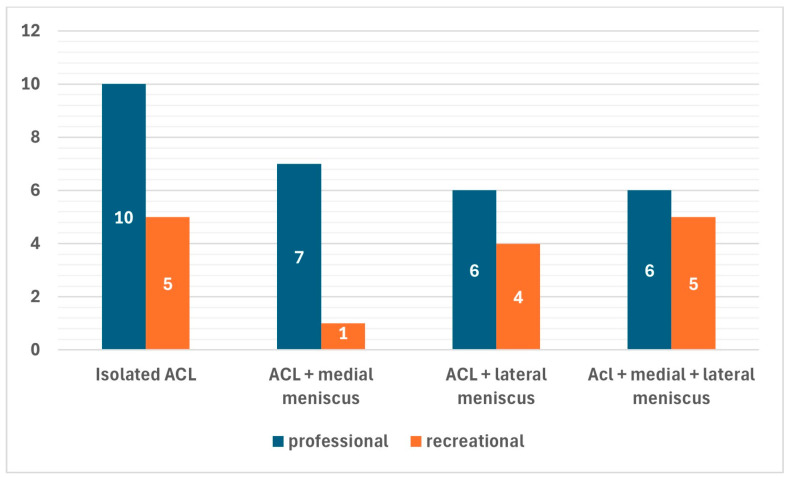
Prevalence of concomitant meniscus tears within the two groups of professional and recreational athletes.

**Figure 4 healthcare-13-01056-f004:**
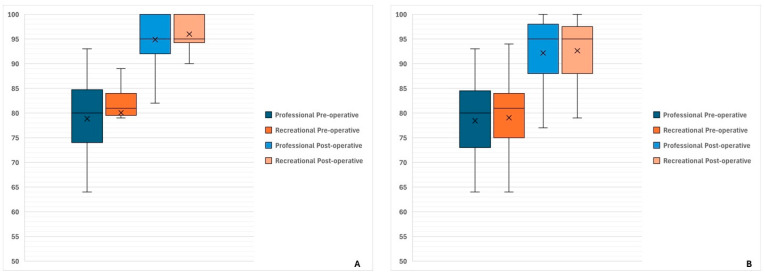
Functional scores at baseline and at last follow-up, with subgroup analysis between professional and recreational athletes. No significant intergroup difference was detected when comparing baseline outcomes as well as final results. Conversely, a statistically significant improvement was found at last follow-up in both groups. (**A**) Lysholm Score. (**B**) International Knee Documentation Committee (IKDC) subjective score.

**Table 1 healthcare-13-01056-t001:** Baseline demographics and surgical details of the sample evaluated in this study. Values are expressed as absolute numbers and related percentages between brackets.

Gender	
Male	7 (15.9%)
Female	37 (84.1%)
Age (years)	21.2 ± 5.0
BMI	22.6 ± 2.8
Follow-up (months)	27.0 ± 12.2
Sport participation	
Professional	29 (65.9%)
Recreational	15 (34.1%)
Graft choice	
Hamstrings	40 (90.9%)
BPTB	3 (6.8%)
Quadriceps tendon	1 (2.3%)
Type of surgical procedure	
Isolated ACL reconstruction	14 (31.8%)
ACL reconstruction + partial meniscectomy	3 (6.8%)
ACL reconstruction + meniscus repair	27 (61.4%)

ACL: anterior cruciate ligament; BTBP: bone–patellar tendon–bone; BMI: body mass index. + means “and”.

**Table 2 healthcare-13-01056-t002:** Return-to-sports rates, clinical examination data, complications and reoperations rates.

	Professional	Recreational	*p*-Values
Return to play	24 (82.8%)	9 (60.0%)	n.s.
Return to pre-injury sport level	23 (79.3%)	6 (40.0%)	0.0091 ^†^
Lachman test *			
Low grade (grade 0–1)	14 (87.4%)	13 (100%)	n.s.
High grade (grade 2–3)	2 (12.6%)	0	n.s.
Pivot shift test *			
Low grade (grade 0–1)	15 (93.8%)	13 (100%)	n.s.
High grade (grade 2–3)	1 (6.2%)	0	n.s.
KT-1000 side-to-side difference *			
Normal (<3 mm)	14 (87.4%)	13 (100%)	n.s.
Higher than normal (≥3 mm and <5 mm)	1 (6.2%)	0	n.s.
Frankly abnormal (≥5 mm)	1 (6.2%)	0	n.s.
Complications	3 (10.3%)	0	n.s.
ACL Failures	2 (6.9%)	0	n.s.
Contralateral ACL ruptures	2 (6.9%)	1 (6.7%)	n.s.
Re-operations	4 (13.8%)	2 (13.3%)	n.s.

* Results from 29 patients available for in-person clinical and instrumental evaluation.^†^ = *p* < 0.05, n.s. = not statistically significant.

## Data Availability

Data are contained within the article.

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
