# Peer review of "Anterior Cruciate Ligament Reconstruction with Modified Transtibial Technique: Outcomes and Return to Sport in Athletes"

_healthcare, 2025, doi:10.3390/healthcare13091056_

Round 1

Reviewer 1 Report

Comments and Suggestions for Authors

Thank you for the opportunity. There are few concerns regarding this well designed and wrtten article.

  1. In the line 134- Is it MCL? It should be medial meniscus.
  2. There were three grafts were used. But the outcome was calculated for all three. I would say the out come for this particular technique should be assessed separately fpr all three grafts. 
  3. If authors going to do the assessments separately for three types of grafts, adequacy of numbers of BTTP and QT are questionable.Thank you again. Best wishes

Author Response

Author's Reply to the Review Report (Reviewer 1)

Thank you for your valuable contribution to the manuscript review process.

Below you will find the revision point by point and the changes in the text marked in red.

Comment 1: In the line 134- Is it MCL? It should be medial meniscus.

Response1: Thank you for your suggestion. We have inserted the correction in the text.

Comment 2: There were three grafts were used. But the outcome was calculated for all three. I would say the outcome for this particular technique should be assessed separately for all three grafts.

Response2: Thank you for your important comment. We insert this bias as a limitation of our study.

Comment 3: If authors going to do the assessments separately for three types of grafts, adequacy of numbers of BTTP and QT are questionable. Thank you again. Best wishes

Response 3: Thank you. Unfortunately, the numbers of BTTP and QT are not an adequate number to evaluate separately the three types of grafts

Reviewer 2 Report

Comments and Suggestions for Authors

The introduction briefly mentions the modified transtibial technique, but its specific differences from the conventional transtibial approach  are not clearly elaborated. It would improve the manuscript to expand on how the modification overcomes limitations of the traditional transtibial method. 

In the section 2.2. Surgical technique and rehabilitation protocol, it would be helpful to include  intraoperative photographs or schematic illustrations demonstrating the key steps or positioning differences of the modified transtibial technique.

Given the impact of concomitant meniscal injuries on return-to-sport outcomes, it would be beneficial to stratify patients into subgroups based on the type of meniscal management (no tear, partial meniscectomy, meniscus repair).

Although the study primarily utilized hamstring autografts (90.9%), a small number of patients received BPTB (n = 3) or quadriceps tendon grafts (n = 1). Given the very limited number of these alternative graft types, their inclusion may introduce variability in outcomes without contributing meaningful subgroup comparisons. Consider excluding these few cases from the final analysis.

Fig 4 did not add value to the current manuscript, especially considering the relatively low event rate and small sample size. The survival analysis results are already clearly presented in the text and Table 2. Consider removing Fig 4.

If the authors decide to include a subgroup analysis based on meniscus management, the Discussion section should be expanded accordingly.

Comments on the Quality of English Language

Manuscript contains minor language issues and some redundancies that should be addressed to improve clarity and fluency.

Author Response

Author's Reply to the Review Report (Reviewer 2)

Thank you for your valuable contribution to the manuscript review process.

Below you will find the revision point by point and the changes in the text marked in red.

Comment 1: The introduction briefly mentions the modified transtibial technique, but its specific differences from the conventional transtibial approach are not clearly elaborated. It would improve the manuscript to expand on how the modification overcomes limitations of the traditional transtibial method.

Response1: Thank you for your valuable suggestion. We have expanded the introduction to more clearly describe the differences between the modified and conventional transtibial techniques. Specifically, we have highlighted how the modified approach allows for more anatomical femoral tunnel placement by adjusting the orientation and position of the tibial tunnel, thereby addressing the limitations associated with the traditional technique. We appreciate your comment, which helped us improve the clarity and depth of our manuscript.

Comment 2: In the section 2.2. Surgical technique and rehabilitation protocol, it would be helpful to include intraoperative photographs or schematic illustrations demonstrating the key steps or positioning differences of the modified transtibial technique.

Response 2: Thank you for your valuable suggestion. I have now included intraoperative photographs in section 2.2, to better illustrate the key steps and positioning differences of the modified transtibial technique.

Comment 3: Given the impact of concomitant meniscal injuries on return-to-sport outcomes, it would be beneficial to stratify patients into subgroups based on the type of meniscal management (no tear, partial meniscectomy, meniscus repair).

Response 3: Thank you for your valuable suggestion. It is much appreciated, and we completely agree that stratifying patients based on the type of meniscal management (no tear, partial meniscectomy, meniscus repair) will provide more meaningful insights into return-to-sport outcomes. We will include this as a potential bias in the limitations of the study, acknowledging its impact on the results.

Comment 4: Although the study primarily utilized hamstring autografts (90.9%), a small number of patients received BPTB (n = 3) or quadriceps tendon grafts (n = 1). Given the very limited number of these alternative graft types, their inclusion may introduce variability in outcomes without contributing meaningful subgroup comparisons. Consider excluding these few cases from the final analysis.

Response 4: Thank you for your valuable observation. We agree that the inclusion of a small number of patients with BPTB or quadriceps tendon grafts may introduce variability in the outcomes. This important potential bias has now been acknowledged and included in the limitations section of the manuscript.

Comment 5: Fig 4 did not add value to the current manuscript, especially considering the relatively low event rate and small sample size. The survival analysis results are already clearly presented in the text and Table 2. Consider removing Fig 4.

Response 5: Thank you for your helpful feedback. We agree that, given the low event rate and limited sample size, Figure 4 may not significantly enhance the presentation of the survival analysis results, which are already clearly reported in the text and Table 2. As suggested, we have removed Figure 4 from the manuscript to improve clarity and conciseness.

Comment 6: If the authors decide to include a subgroup analysis based on meniscus management, the Discussion section should be expanded accordingly.

Response 6: Thank you for your suggestion. While we recognize the relevance of a subgroup analysis based on meniscus management, due to the retrospective nature of the study and limited sample size within subgroups, we have chosen not to perform a full stratified analysis. However, we have acknowledged this as a limitation in the revised manuscript.

Comment 7: Manuscript contains minor language issues and some redundancies that should be addressed to improve clarity and fluency.

Response 7: Thank you for your comment. We have had the manuscript professionally edited for language to address the minor issues and improve clarity and fluency. A language editing certificate has been included in the supplementary files for your reference.

Reviewer 3 Report

Comments and Suggestions for Authors

General comments:

The manuscript entitled "Anterior Cruciate Ligament Reconstruction using a Modified Transtibial Technique in Recreational and Professional Athletes: Clinical Outcomes and Return to Sport at a Minimum Follow-up of 12 Months" presents a retrospective analysis of ACL reconstructions performed using a modified transtibial technique. The study examines functional outcomes and return to sport rates in both professional and recreational athletes.

The topic is clinically relevant and the authors have presented the data clearly. However, the manuscript has several important limitations that need to be addressed before it can be considered for publication. The lack of a control group, the limited sample size, the relatively short follow-up, and the lack of methodological detail regarding the surgical technique reduce the scientific impact and generalizability of the findings. Furthermore, although the data confirm existing knowledge, the manuscript lacks significant novelty at this time.

Major concerns

  1.    Lack of a control group

 1) The lack of a control group using a different ACL reconstruction technique (e.g. anteromedial portal or outside-in) is a major limitation. Without such a comparison, conclusions about the efficacy or superiority of the modified transtibial technique remain speculative.

2) A more cautious tone should be adopted when interpreting the results.

2.   Insufficient description of the surgical technique

1) The modified transtibial technique is the main focus of the manuscript, but its detailed description is lacking.

2) The authors should include a schematic diagram or intraoperative photograph to visually explain the unique features of the technique and how it differs from conventional transtibial or independent drilling methods.

3. Limited sample size and follow-up

1) The study includes only 44 patients with a mean follow-up of 27 months, which is insufficient to assess long-term outcomes such as graft durability or osteoarthritis progression.

2) The small sample size limits the statistical power, especially for subgroup analyses.

4.  Inconsistent follow-up methods

 1) Clinical outcomes were assessed by both personal examination (n=29) and telephone interview (n=15). This heterogeneity in assessment methods may affect the reliability of the results, particularly with regard to the assessment of knee stability.

2) Consider analysing and presenting the two groups separately or acknowledge this limitation more explicitly.

5. Lack of novelty

1) Many of the findings (e.g. functional improvement, return to play rates, higher RTP in professionals) are already well documented in the literature.

2) The manuscript would benefit from a clearer statement of its unique contribution to current knowledge.

6.  Unclear criteria for LET (lateral extra-articular tenodesis)

1) Only 3 cases received LET, but the indications for its use remain vague ("at surgeon's discretion").

2) Please provide more standardized criteria (e.g., pivot-shift grade, graft size, patient age or sport level).

Minor Concerns

  1. Definition of "Return to Sport"

1)The definition of "return to sport" and "preinjury level" should be clearly stated. Was the Tegner score used to verify these outcomes?

2. Selection Bias and Ethical Approval Timing

1) The ethical approval date (October 2024) appears to postdate the inclusion period (2020–2023). Please clarify whether retrospective approval was obtained.

  1. Gender and Sport Distribution

1)The majority of the cohort were male soccer players. Please discuss how this population bias may affect the generalizability of your findings.

4. Statistical Analysis

1) Please clarify if adjustments were made for multiple comparisons, especially in subgroup analyses.

5. Language and Clarity

1)The manuscript would benefit from careful English language editing to improve fluency and clarity.

Comments on the Quality of English Language

The manuscript would benefit from careful English language editing to improve fluency and clarity.

Author Response

Author's Reply to the Review Report (Reviewer 3)

Thank you for your valuable contribution to the manuscript review process.

Below you will find the revision point by point and the changes in the text marked in red.

Major concerns:

  1. Lack of a control group

Comment 1: The lack of a control group using a different ACL reconstruction technique (e.g. anteromedial portal or outside-in) is a major limitation. Without such a comparison, conclusions about the efficacy or superiority of the modified transtibial technique remain speculative.

Response1: Thank you for this important observation. We fully acknowledge that the absence of a control group using an alternative ACL reconstruction technique (such as anteromedial portal or outside-in drilling) represents a major limitation of the study. As suggested, we have addressed this point in the limitations section of the revised manuscript, emphasizing that, without a direct comparison, conclusions regarding the efficacy or superiority of the modified transtibial technique remain speculative.

Comment 2: A more cautious tone should be adopted when interpreting the results.

Response 2: we have revised the Discussion and Conclusion sections to adopt a more cautious tone when interpreting the results, especially in light of the study's limitations. We now emphasize that the findings should be interpreted within the context of the study design and the absence of a comparative control group.

  1. Insufficient description of the surgical technique

Comment 3: The modified transtibial technique is the main focus of the manuscript, but its detailed description is lacking.

Response 3: We have now expanded the description of the modified transtibial technique in the revised manuscript, providing additional details regarding tunnel placement, surgical steps, and positioning differences compared to the traditional transtibial approach.

Comment 4: The authors should include a schematic diagram or intraoperative photograph to visually explain the unique features of the technique and how it differs from conventional transtibial or independent drilling methods.

Response 4: Thank you for your insightful suggestion. We agree that a visual representation would enhance the reader’s understanding of the modified transtibial technique. In response, we have included a schematic figure that highlights the key steps and anatomical landmarks of the technique and illustrates how it differs from the conventional transtibial and independent drilling methods. We believe this addition will improve the clarity and educational value of the manuscript.

  1. Limited sample size and follow-up

Comment 5: The study includes only 44 patients with a mean follow-up of 27 months, which is insufficient to assess long-term outcomes such as graft durability or osteoarthritis progression.

Response 5: Thank you for your valuable comment. We fully acknowledge that the relatively small sample size and the mean follow-up of 27 months represent limitations of the study, particularly in evaluating long-term outcomes such as graft durability or osteoarthritis progression. This limitation has been clearly addressed in the revised manuscript under the limitations section. Future studies with larger cohorts and longer follow-up periods are needed to confirm our findings and better assess long-term outcomes.

Comment 6: The small sample size limits the statistical power, especially for subgroup analyses.

Response 6: Thank you for your observation. We agree that the small sample size limits the statistical power of the study, particularly in relation to subgroup analyses. This limitation has been acknowledged in the revised manuscript, and we have interpreted the subgroup findings with appropriate caution.

  1. Inconsistent follow-up methods

Comment 7: Clinical outcomes were assessed by both personal examination (n=29) and telephone interview (n=15). This heterogeneity in assessment methods may affect the reliability of the results, particularly with regard to the assessment of knee stability.

Response 7: Thank you for your valuable comment. We acknowledge that the use of both personal examination (n=29) and telephone interviews (n=15) to assess clinical outcomes introduces a potential source of heterogeneity, particularly when evaluating knee stability. In the revised manuscript, we have discussed this limitation and emphasized the need for a more uniform approach to outcome assessment in future studies to ensure greater reliability.

Comment 8: Consider analysing and presenting the two groups separately or acknowledge this limitation more explicitly.

Response 8: We have decided to explicitly acknowledge this limitation in the revised manuscript. Additionally, we have discussed the potential impact this may have on the reliability of the results, particularly regarding knee stability.

  1. Lack of novelty

Comment 9: Many of the findings (e.g. functional improvement, return to play rates, higher RTP in professionals) are already well documented in the literature.

Response 9: Thank you for your valuable comment. We acknowledge that many of the findings, such as functional improvement, return-to-play rates, and higher return-to-play rates in professional athletes, are indeed well documented in the existing literature. However, our study provides valuable additional insights by specifically focusing on athletes’ patients. We have revised the manuscript to better highlight the novel aspects of our findings and their potential contributions to the current body of knowledge.

Comment 10: The manuscript would benefit from a clearer statement of its unique contribution to current knowledge.

Response 10: Thank you for your constructive feedback. We appreciate your suggestion and have revised the manuscript to highlight the unique contributions of our study more explicitly.

  1. Unclear criteria for LET (lateral extra-articular tenodesis)

Comment 11: Only 3 cases received LET, but the indications for its use remain vague ("at surgeon's discretion").

Response 11: Thank you for your comment. We acknowledge that the indication for lateral extra-articular tenodesis (LET) in our study was based on the surgeon's discretion, and we agree that this approach may appear somewhat vague. In the revised manuscript, we have clarified the specific criteria used by the surgeons in determining the need for LET, such as cases of high-grade pivot shift, graft size smaller than 8 mm, and signs of constitutional hyperlaxity. We hope this provides greater transparency regarding the decision-making process for this procedure.

Comment 12: Please provide more standardized criteria (e.g., pivot-shift grade, graft size, patient age or sport level).

Response 12: We appreciate the need for more standardized criteria when indicating lateral extra-articular tenodesis (LET). In the revised manuscript, we have provided clearer indication for the use of LET, specifically indicating its use in cases with a high-grade pivot-shift (Grade III), graft diameter smaller than 8 mm, patients under 30 years old, and those engaged in high-demand sports. This should help provide more consistency and clarity regarding the selection criteria.

Minor Concerns:

  1. Definition of "Return to Sport"

Comment 13: The definition of "return to sport" and "preinjury level" should be clearly stated. Was the Tegner score used to verify these outcomes?

Response 13: Thank you for your valuable comment. In the revised manuscript, we have clarified the definitions of "return to sport" and "preinjury level."  The Tegner activity scale was used to quantify activity levels pre- and postoperatively and to verify return to the preinjury level.

  1. Selection Bias and Ethical Approval Timing

Comment 14: The ethical approval date (October 2024) appears to postdate the inclusion period (2020–2023). Please clarify whether retrospective approval was obtained.

Response 14: Thank you for pointing out this important detail. We confirm that the study received retrospective ethical approval in October 2024, as the data collection was performed on previously acquired clinical records from 2020 to 2023. This has been clarified in the revised manuscript to avoid any misunderstanding.

  1. Gender and Sport Distribution

Comment 15: The majority of the cohort were male soccer players. Please discuss how this population bias may affect the generalizability of your findings.

Response 15: Thank you for your thoughtful observation. We acknowledge that the predominance of male soccer players in our cohort introduces a population bias that may limit the generalizability of our findings to other populations, such as female athletes or individuals practicing different sports. This limitation has been addressed in the revised manuscript, and we have added a statement in the discussion to clarify that caution should be exercised when extrapolating the results to broader athletic or non-athletic populations.

  1. Statistical Analysis

Comment 16: Please clarify if adjustments were made for multiple comparisons, especially in subgroup analyses.

Response 16: We confirm that no statistical adjustments for multiple comparisons were performed in this study, due to the exploratory nature of the subgroup analyses and the limited sample size. This limitation has been acknowledged in the revised manuscript to ensure transparency and to caution against overinterpretation of subgroup findings.

  1. Language and Clarity

Comment 17: The manuscript would benefit from careful English language editing to improve fluency and clarity.

Response 17: Thank you for your comment. The manuscript has undergone thorough English language editing by a native speaker to improve fluency and clarity. A language editing certificate has been included in the supplementary files for your review.

Reviewer 4 Report

Comments and Suggestions for Authors

Title

Anterior Cruciate Ligament Reconstruction using a modified Transtibial Technique in recreational and professional Athletes Clinical Outcomes and Return to Sport after a Minimum Follow-up of 12 months

Reviewer comments:

Can you please make the title within 15 words range?

General comments:

Thank you very much for giving me this opportunity to read this manuscript. The manuscript requires a general review for writing style and grammar. For example: redundant phrasing (e.g., "concomitant meniscus tears" repeated frequently) and "Athletic athletes" is redundant.

Abstract:

  • General: The notion of “Return to preinjury level" is a bit subjective. I am wondering if this self-reported or objectively assessed. Please consider clarifying this point in the abstract.
  • General: Confidence intervals (CI) are missing for key outcomes (e.g., failure rate). Please consider inserting them in the Results part.

Introduction:

  • P2, L58-59: It seems that the transition to "anatomic ACL reconstruction as the gold standard" is rushed. Missing a brief critique of why independent drilling techniques (AM portal/outside-in) became preferred over traditional transtibial. Can you please provide more details to the reader with supporting references?
  • P2, L63: It seems that the supposed advantages (e.g., "suboptimal graft orientation with lower bending angle") are somewhat unclear and seemingly contradictory ("suboptimal" sounds negative, but the text suggests benefits). Can you please consider rewriting this paragraph to enhance clarity and prevent confusions?
  • General: The introduction is a bit short. Furthermore, the gap in knowledge is somewhat vague. Phrases such as "questions remain" and "outcomes can vary" don’t sharply define what’s missing in the literature. I recommend explaining to the reader further in one or two paragraphs the importance of this current study.
  • P2, L75-76: After reading the introduction several times, I believe that the hypothesis might be overly broad ("good clinical outcomes and high RTS rates"). It should be testable and specific. Can you please consider this and ensure that the hypothesis is more specific?

Materials and Methods:

  • P3,L89: "Athletic athletes" is redundant. Exclusion criteria could be more precise.
  • General: It seems that Professional vs. recreational definition is arbitrary (income-based). Please consider defining them more appropriately.
  • P3, L114-115: Missing justification for graft selection (surgeon preference is vague). Can you please add more details?
  • P3, L116-118: Please provide a reference to the following paragraph “This concomitant procedure was added at the surgeons’ discretion, usually in cases of graft size less than 8 mm, high-grade pivot-shift without concomitant lateral meniscus tears and/or signs of constitutional hyperlaxity.”
  • P3, L117: “graft size <8 mm" – Can you please explain to the reader why this threshold?
  • General: Please be careful, variable brace duration was between 3–6 weeks and RTS timeline was between 5–6 months and those lack rationales. Please consider adding more clarifications.
  • P4, L167: Phone interviews for functional scores may reduce accuracy. Please consider this in the limitations of this study in the discussion section.
  • General: No power analysis for sample size justification. Can you please add a paragraph that explain how sample size was calculated?
  • General: Please add more details about sampling methods.

Results:

  • General: Missing confidence intervals (CIs) for key outcomes (e.g., failure rate = 4.6%).
  • General: Please be careful, it seems that no data on Tegner scores pre- vs. post-injury were displayed in the result section regardless to the fact that they were mentioned in Methods but not reported in the result section. Please add more details.
  • General: Phone follow-up cohort (34.1%) may bias functional scores. Please consider including this within the limitations subsection.

Discussion:

  • General: The discussion jumps between topics (failure rates → meniscus → RTS → limitations) without clear transitions. Please consider using sign posting while revising the discussion section.
  • General: Please be careful, there were few claims of "satisfactory outcomes" and "low failure rates", which lack direct comparison to other techniques. Please consider rewriting them or provide supporting statements and references.
  • General: there an overrepresentation of male gender and no mention of gender bias (84% male cohort) in the limitations subsection. Please consider rewriting the limitations subsection according to the above recommendations and suggestions above.

Conclusion

  • Please consider rewriting the conclusion to reflect all amendments after reviewers’ comments and suggestions.

Good luck

Comments on the Quality of English Language

The manuscript requires a general review for writing style and grammar. For example: redundant phrasing (e.g., "concomitant meniscus tears" repeated frequently) and "Athletic athletes" is redundant.

Author Response

Author's Reply to the Review Report (Reviewer 4)

Thank you for your valuable contribution to the manuscript review process.

Below you will find the revision point by point and the changes in the text marked in red.

Title:

Comment 1: Can you please make the title within 15 words range?

Response 1: Thank you for your helpful suggestion. As requested, we have revised the title of the manuscript to make it more concise and within the 15-word limit.

General comments:

Comment 2: Thank you very much for giving me this opportunity to read this manuscript. The manuscript requires a general review for writing style and grammar. For example: redundant phrasing (e.g., "concomitant meniscus tears" repeated frequently) and "Athletic athletes" is redundant.

Response 2: Thank you very much for your valuable feedback and for the opportunity to improve our manuscript. We have carefully revised the text to correct issues related to writing style and grammar. Redundant phrasing, such as "concomitant meniscus tears" and "athletic athletes," has been removed or reworded for clarity and precision. We have had the manuscript professionally edited for language to address the minor issues and improve clarity and fluency. A language editing certificate has been included in the supplementary files for your reference.

Abstract:

Comment 3: General: The notion of “Return to preinjury level" is a bit subjective. I am wondering if this self-reported or objectively assessed. Please consider clarifying this point in the abstract.

Response 3: Thank you for your valuable comment. In the revised manuscript, we have clarified the definitions of "return to sport" and "preinjury level."  The Tegner activity scale was used to quantify activity levels pre- and postoperatively and to verify return to the preinjury level.

Comment 4: General: Confidence intervals (CI) are missing for key outcomes (e.g., failure rate). Please consider inserting them in the Results part.

Response 4: Thank you for your helpful observation. We agree that including confidence intervals (CI) adds valuable information regarding the precision of key outcomes.

Introduction:

Comment 5: P2, L58-59: It seems that the transition to "anatomic ACL reconstruction as the gold standard" is rushed. Missing a brief critique of why independent drilling techniques (AM portal/outside-in) became preferred over traditional transtibial. Can you please provide more details to the reader with supporting references?

Response 5: Thank you for your insightful comment. We agree that the transition to anatomic ACL reconstruction merits further clarification. We have revised the manuscript to include a brief critique of the traditional transtibial technique and to explain why independent drilling techniques (anteromedial portal and outside-in) have become preferred.

Comment 6: P2, L63: It seems that the supposed advantages (e.g., "suboptimal graft orientation with lower bending angle") are somewhat unclear and seemingly contradictory ("suboptimal" sounds negative, but the text suggests benefits). Can you please consider rewriting this paragraph to enhance clarity and prevent confusions?

Response 6: Thank you for your valuable feedback. We appreciate your observation and agree that the wording in the original paragraph may have caused confusion. In response, we have revised the text to clarify that while certain graft characteristics have traditionally been viewed as suboptimal, some studies suggest they may offer biomechanical advantages in specific situations.

Comment 7: General: The introduction is a bit short. Furthermore, the gap in knowledge is somewhat vague. Phrases such as "questions remain" and "outcomes can vary" don’t sharply define what’s missing in the literature. I recommend explaining to the reader further in one or two paragraphs the importance of this current study.

Response7: Thank you for your thoughtful feedback. We acknowledge that the original introduction was concise and that the gap in knowledge could have been more clearly articulated. In response, we have expanded the introduction by adding one additional paragraph to better define the current limitations in the literature and to more explicitly state the rationale for this study. Specifically, we now emphasize the lack of consensus on the clinical implications of femoral tunnel positioning and graft orientation in ACL reconstruction, despite advances in anatomic techniques. We also highlight the need for further comparative studies to assess how different surgical approaches influence long-term outcomes. We hope these additions provide a clearer justification for the relevance and significance of our work.

Comment 8: P2, L75-76: After reading the introduction several times, I believe that the hypothesis might be overly broad ("good clinical outcomes and high RTS rates"). It should be testable and specific. Can you please consider this and ensure that the hypothesis is more specific?

Response 8: Thank you for your valuable feedback. We understand that the initial hypothesis may have been too broad in its formulation. To ensure that the hypothesis is both specific and testable, we have revised it to focus on a clearly defined intervention and measurable clinical outcomes.

Materials and Methods:

Comment 9: P3,L89: "Athletic athletes" is redundant. Exclusion criteria could be more precise.

Response 9: Thank you for your comment. We have revised the text to remove the redundant wording ("athletic athletes") and have rewritten the exclusion criteria to make them more precise and specific.

Comment 10: General: It seems that Professional vs. recreational definition is arbitrary (income-based). Please consider defining them more appropriately.

Response 10:  Thank you for your valuable feedback. We have revised the manuscript to provide a more detailed and appropriate definition of professional and recreational athletes.

Comment 11: P3, L114-115: Missing justification for graft selection (surgeon preference is vague). Can you please add more details?

Response 11: Thank you for your observation. We have updated the manuscript to provide a more detailed explanation regarding graft selection. The choice of graft was primarily based on a combination of patient-specific factors (e.g., age, sport type, level of competition) and intraoperative findings, in addition to surgeon experience and familiarity with specific techniques. These considerations have now been clarified in the text.

Comment 12: P3, L116-118: Please provide a reference to the following paragraph “This concomitant procedure was added at the surgeons’ discretion, usually in cases of graft size less than 8 mm, high-grade pivot-shift without concomitant lateral meniscus tears and/or signs of constitutional hyperlaxity.”

Response 12: We have included a reference to support the paragraph, as requested.

Comment 13: P3, L117: “graft size <8 mm" – Can you please explain to the reader why this threshold?

Response 13: We have added an explanation to clarify the rationale behind the 8 mm graft size threshold.

Comment 14: General: Please be careful, variable brace duration was between 3–6 weeks and RTS timeline was between 5–6 months and those lack rationales. Please consider adding more clarifications.

Response 14: Thank you for the feedback. We have revised the text to include clarifications regarding both the variable brace duration and the return-to-sport (RTS) timeline.

Comment 15: P4, L167: Phone interviews for functional scores may reduce accuracy. Please consider this in the limitations of this study in the discussion section.

Response 15: Thank you for the comment. We have acknowledged the potential limitation of using phone interviews for collecting functional scores and have included this point in the limitations section of the Discussion.

Comment 16: General: No power analysis for sample size justification. Can you please add a paragraph that explain how sample size was calculated?

Response 16: We have added the requested paragraph on sample size justification to the Methods section.

Comment 17: General: Please add more details about sampling methods.

Response 17: Thank you for the comment. We have added further details regarding the sampling methods to clarify how participants were selected.

Results

Comment 18: General: Missing confidence intervals (CIs) for key outcomes (e.g., failure rate = 4.6%).

Response 18: Thank you for your helpful observation. We agree that including confidence intervals (CI) adds valuable information regarding the precision of key outcomes.

Comment 19: General: Please be careful, it seems that no data on Tegner scores pre- vs. post-injury were displayed in the result section regardless to the fact that they were mentioned in Methods but not reported in the result section. Please add more details.

Response 19: Thank you for pointing this out. We have revised the Results section to include the Tegner scores, as they were described in the Methods section but previously omitted from the results.

Comment 20: General: Phone follow-up cohort (34.1%) may bias functional scores. Please consider including this within the limitations subsection.

Response 20: Thank you for the comment. We have added a discussion regarding the potential bias introduced by the phone follow-up cohort in the limitations section.

Discussion:

Comment 21: General: The discussion jumps between topics (failure rates → meniscus → RTS → limitations) without clear transitions. Please consider using sign posting while revising the discussion section.

Response 21: Thank you for the helpful feedback. We have revised the Discussion section to improve clarity and flow by adding signposting and organizing the content into clearly defined thematic subsections.

Comment 22: General: Please be careful, there were few claims of "satisfactory outcomes" and "low failure rates", which lack direct comparison to other techniques. Please consider rewriting them or provide supporting statements and references.

Response 22: Thank you for your valuable comment. We have revised the claims of “satisfactory outcomes” and “low failure rates” to either provide appropriate references or rephrase them to reflect a more objective tone.

Comment 23: General: there an overrepresentation of male gender and no mention of gender bias (84% male cohort) in the limitations subsection. Please consider rewriting the limitations subsection according to the above recommendations and suggestions above.

Response 23: Thank you for your valuable comment. We fully acknowledge the gender imbalance in our cohort, with 84.1% of the included athletes being male. We have revised the Limitations subsection to explicitly address this issue, noting that the overrepresentation of male participants may limit the generalizability of our findings to female athletes.

Conclusion

Comment 24: Please consider rewriting the conclusion to reflect all amendments after reviewers’ comments and suggestions.

Response 24: Thank you for your suggestion. We have carefully revised the Conclusion section to reflect all amendments made throughout the manuscript following the reviewers’ comments.

Quality of English Language

Comment 25: The manuscript requires a general review for writing style and grammar. For example: redundant phrasing (e.g., "concomitant meniscus tears" repeated frequently) and "Athletic athletes" is redundant.

Response 25: Thank you for your valuable feedback. We have carefully revised the text to correct issues related to writing style and grammar. Redundant phrasing, such as "concomitant meniscus tears" and "athletic athletes," has been removed or reworded for clarity and precision. We have had the manuscript professionally edited for language to address the minor issues and improve clarity and fluency. A language editing certificate has been included in the supplementary files for your reference.

Round 2

Reviewer 3 Report

Comments and Suggestions for Authors

Thank you for your careful and thorough revisions.

You have successfully addressed all major and minor concerns raised during the review process.
In particular, your expanded description of the surgical technique, the addition of a schematic figure, the clarification of the indications for LET, and the more cautious interpretation of the study results have significantly improved the clarity, transparency, and scientific rigor of the manuscript.

I have no further major concerns, and I believe your revised manuscript is now suitable for publication.